# Nuclear export is a limiting factor in eukaryotic mRNA metabolism

**Jason M. Müller**[1,2☉], **Katharina Moos**[2,3☉], **Till Baar**[2], **Kerstin C. Maier**[4], **Kristina Zumer**[4]*, **Achim Tresch**[1,2,5]*

**1** Cologne Excellence Cluster on Cellular Stress Responses in Aging-Associated Diseases (CECAD), University of Cologne, Cologne, Germany, **2** Institute of Medical Statistics and Computational Biology, Faculty of Medicine, University of Cologne, Cologne, Germany, **3** Institute of Medical Bioinformatics and Systems Medicine, Medical Center - University of Freiburg, Faculty of Medicine, University of Freiburg, Freiburg, Germany, **4** Department of Molecular Biology, Max Planck Institute for Multidisciplinary Sciences, Göttingen, Germany, **5** Center for Data and Simulation Science, University of Cologne, Cologne, Germany

☉ These authors contributed equally to this work.
* kristina.zumer@mpinat.mpg.de (KZ); achim.tresch@uni-koeln.de (AT)

**Data Availability Statement:** Raw and processed sequencing data used in this work has been deposited in NCBI's Gene Expression Omnibus and are accessible through GEO Series accession number GSE233546. The source code associated

## Abstract

The eukaryotic mRNA life cycle includes transcription, nuclear mRNA export and degradation. To quantify all these processes simultaneously, we perform thiol-linked alkylation after metabolic labeling of RNA with 4-thiouridine (4sU), followed by sequencing of RNA (SLAM-seq) in the nuclear and cytosolic compartments of human cancer cells. We develop a model that reliably quantifies mRNA-specific synthesis, nuclear export, and nuclear and cytosolic degradation rates on a genome-wide scale. We find that nuclear degradation of polyadenylated mRNA is negligible and nuclear mRNA export is slow, while cytosolic mRNA degradation is comparatively fast. Consequently, an mRNA molecule generally spends most of its life in the nucleus. We also observe large differences in the nuclear export rates of different 3'UTR transcript isoforms. Furthermore, we identify genes whose expression is abruptly induced upon metabolic labeling. These transcripts are exported substantially faster than average mRNAs, suggesting the existence of alternative export pathways. Our results highlight nuclear mRNA export as a limiting factor in mRNA metabolism and gene regulation.

## Author summary

In our work, we aim to quantify two fundamental processes in the life cycle of a eukaryotic messenger RNA (mRNA), namely export from the nuclear compartment to the cytosol, and degradation. To enable a genome-wide evaluation, we have implemented an experimental-bioinformatics approach. We use a chemical moiety, 4-thiouridine (4sU), which is incorporated into RNA molecules when added to the solution, to mark RNA which is recently synthesized and distinguish it from already existing RNA. We take a time series of recent and already existing RNA within both the nuclear and cytoplasmic cellular compartments. We have devised a computational model that can reliably quantify mRNA nuclear export and cytosolic degradation rates from this data. We revealed that the export of mRNA from the nucleus to the cytoplasm emerges as a comparatively tardy event.

with this manuscript is available on a GitHub repository at https://github.com/IMSBCompBio/mRNAdynamics.

**Funding:** JMM was funded in parts by the the Cologne Graduate School of Ageing Research (CGA; https://www.ageing-grad-school.de/home). KM was funded in part by an FR2464/4-1 grant from the Deutsche Forschungsgesellschaft (https://www.dfg.de/de). The funders did not play any role in the study design, data collection and analysis, decision to publish, or preparation of the manuscript.

**Competing interests:** The authors have declared that no competing interests exist.

Subsequently, mRNA molecules within the cytoplasm experience swift degradation. This identifies the nucleus as the predominant residence for the greater portion of an mRNA's lifecycle. Yet, we have also found notable exceptions from this scheme which are immediately exported after transcription without delay, suggesting the plausible existence of alternative mRNA export pathways. In sum, our research underscores the pivotal role of nuclear mRNA export as a determining factor in the orchestration of mRNA metabolism and the regulation of gene expression within eukaryotic cells.

## Introduction

In brief, the life cycle of an mRNA consists of its transcription, translation and degradation. Gene regulatory mechanisms are intervening in all three processes. In eukaryotic cells, the processes of transcription and translation are spatially separated into the nucleus and cytoplasm, respectively. The export of mRNA to the cytosol adds an additional layer of regulation to RNA metabolism, which will be investigated in this work. Differences in nuclear RNA export will affect cytoplasmic mRNA levels and thus the availability of RNA for translation. For example, slow RNA export under steady-state conditions could buffer the variation in nuclear RNA abundance caused by transcriptional bursting, leading to stable cytoplasmic mRNA levels [1, 2]. Dysregulation of mRNA export has also been implicated in several neurodegenerative diseases [3].

To be ready for export, a transcript must be synthesized and released from the transcribing RNA polymerase II. The pre-mRNA undergoes co- and post-transcriptional processing, including 5'-end capping, splicing, 3'-end processing, and association with RNA-binding proteins (RBPs) to form the messenger ribonucleoprotein (mRNP) complex. Each of these maturation steps contributes to the availability of a transcript for export out of the nucleus. Transcripts that are incompletely or incorrectly processed do not pass quality control checkpoints and are either retained in the nucleus until processing is complete or targeted for degradation by the exosome (reviewed in [4]). In contrast to RNA synthesis, RNA degradation occurs in both the cytoplasm and the nucleus. Mature mRNA is mainly degraded in the cytoplasm after deadenylation [5, 6] or microRNA binding (reviewed in [7]). Other RNAs, in contrast, are degraded primarily in the nucleus, where nuclear transcriptome surveillance pathways capture unstable transcripts such as enhancer RNAs, products of pervasive transcription, and improperly processed transcripts [4]. Export of mRNA and non-coding RNA (tRNA, snoRNA, rRNA) occurs via different pathways (reviewed in [8]). Several adaptor proteins mediate the export of a mature mRNP through the nuclear pore complex [9]. In contrast, improperly processed mRNAs do not form export competent mRNPs, are therefore retained in the nucleus and targeted for degradation [10–12]. The export dynamics of certain RNAs differ due to the use of non-canonical adaptor proteins or the tethering of the genomic locus to the nuclear pore [13–15].

mRNA export has primarily been measured by RNA Fluorescence In Situ Hybridization (RNA-FISH) for individual transcripts [16–18] and has also been scaled up to obtain export estimates for several hundred transcripts [19]. Genome-wide export estimates have been obtained by fractionation of cells combined with sequencing of total RNA in each fraction [1]. More recently, subcellular fractionation was combined with metabolic labeling of RNA and isolation of labeled RNA to quantify newly synthesized RNA in different compartments of *Drosophila* cells [20]. In contrast to monitoring mRNA decay upon transcriptional inhibition [21], radioactive or metabolic labeling of newly synthesized RNA allows monitoring of RNA

metabolism [22, 23]. The use of nucleoside-analogs such as 4-thiouridine (4sU) have been considered as minimally perturbing and has also been applied to monitor mRNA metabolism in single cells [2, 24]. Short RNA labeling pulses are suitable for measuring RNA synthesis rates, while longer pulses are better suited for estimating RNA degradation rates [25]. Statistical models that complement these experimental approaches employ a single-compartment model assuming constant RNA synthesis rates, degradation rates, and a dynamic equilibrium of each RNA population [26]. While these single-compartment models give insight into the overall dynamics of an RNA within a cell, it does not provide insights into subcellular dynamics such as the speed of RNA export.

Previous studies of RNA metabolism have relied on perturbation of transcription or isolation of newly synthesized RNA, which is a multi-step procedure leading to higher technical variation [27, 28]. To overcome the limitations of the technical variation due to isolation of newly synthesized RNA, we used thiol-linked alkylation for metabolic sequencing of RNA (SLAM-seq) [29, 30]. SLAM-seq applies labeling of newly synthesized RNA with 4sU and thiol-linked alkylation to induce T>C transitions at positions where 4sU was incorporated. These conversions can be detected by sequencing without prior isolation of the labeled fraction. We combined this technology with subcellular fractionation to monitor RNA metabolism in the nuclear and cytoplasmic fraction simultaneously. To that end, we have developed a probablistic framework to reliably fit a two-compartment model of RNA metabolism. We compute RNA synthesis and nuclear export rates, as well as nuclear and cytosolic RNA degradation rates on a genome-wide scale which overcomes some limitations of a single-compartment model. Our estimates confirm that nuclear degradation of polyadenylated mRNA is negligible, and mRNA export is slow compared to its cytosolic degradation. Consequently, mRNA remains in the nucleus for considerably longer than in the cytosol. We are also able to quantify the metabolism of alternative 3' untranslated region (UTR) transcript isoforms of genes. Variation in 3'UTR length alters RBP binding and is known to affect transcript stability or localization [31–33]. We show that different 3'UTR transcript isoforms can be exported at substantially different rates and that 3'UTR length is a relevant factor. Finally, we discover genes whose synthesis is abruptly induced upon 4sU labeling and whose export is an order of magnitude faster than that of average mRNAs, pointing to the existence of stress-induced alternative export pathways.

## Results

### SLAM-seq time-series and 3'UTR quantification

We have performed a SLAM-seq time-series experiment in HeLa-S3 cells, where two replicate samples were taken at t = 0, 15, 30, 45, 60, 90, 120 and 180 min after the addition of 500 $\mu$M 4sU (Fig 1A and A in S1 Appendix). After metabolic labeling of RNA, the cells were fractionated to obtain the nuclear and the cytosolic RNA fractions. The accuracy of the subcellular fractionation was validated by Western Blots (Section 1 and Fig B in S1 Appendix).

To quantify mRNA metabolism, polyadenylated transcripts were captured by targeted sequencing library preparation and sequencing of polyadenylated 3' ends (3'-seq). Reads were mapped to the human genome using SlamDunk [34], yielding between 8.5–24.4 million uniquely mapped reads per sample (Section 2–3 in S1 Appendix). The mapping efficiency ranged from 66% and 81%. To exclude a mapping bias against labeled reads, we extensively investigated the reasons for dropouts (Section 2–3 and Figs C-D in S1 Appendix).

Next, to analyse mRNA metabolism at the transcript level, we merged all reads pertaining to annotated 3'UTRs of a gene. A fraction of 61% to 78% of all uniquely mapped reads per sample aligned to 3'UTRs (Section 2–3 and Fig E in S1 Appendix). After merging overlapping

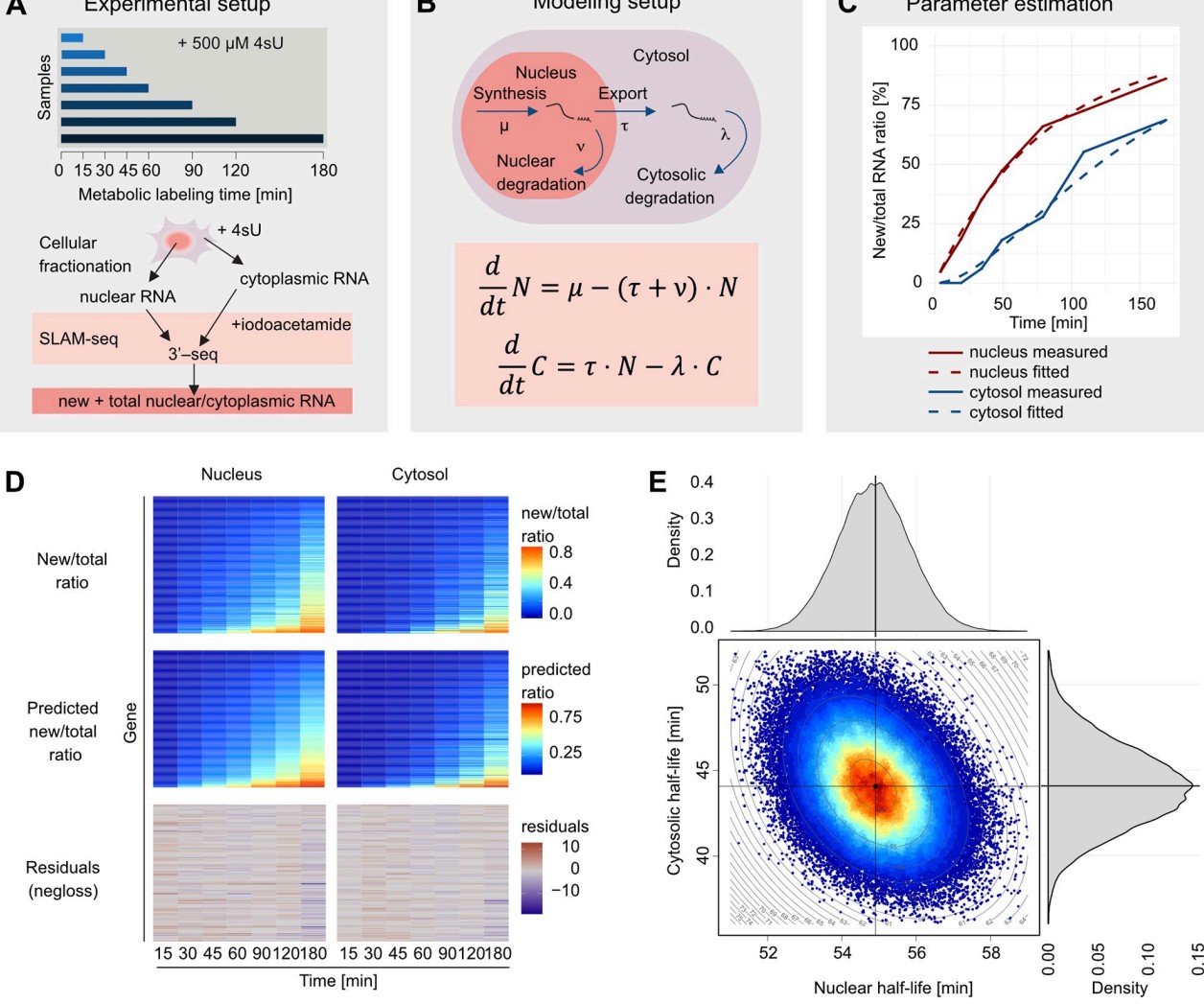

**Fig 1. Experimental setup and computational modeling of RNA metabolism.** (A) Schematic representation of the experimental setup. SLAM-seq time series samples were generated from fractionated nuclear and the cytoplasmic fractions at $t$ = 0, 15, 30, 45, 60, 90, 120, 180 min after addition of $500\mu M$ 4sU and preprocessed to obtain new and total reads. (B) Two compartment differential equation model of the nuclear RNA fraction ($N = N(t)$) and the cytosolic RNA fraction ($C = C(t)$). These fractions are described by 4 parameters, namely the synthesis rate $\mu$, the nuclear degradation rate $\nu$, the nuclear export rate $\tau$, and the cytosolic degradation rate $\lambda$. (C) Parameter fitting of the nuclear removal rate $\nu + \tau$ and the cytosolic degradation rate $\lambda$ at the example of the 3'UTR of the MAFG gene using its new by total RNA ratios. (D) Measured $\frac{new}{total}$ RNA ratios, estimated $\frac{new}{total}$ RNA ratios and the respective residuals after parameter fitting for the nuclear (left) and cytoplasmic (right) fractions. Each row corresponds to one 3'UTR with reliable parameter estimates. (E) Heat scatterplot of a 2-dimensional MCMC sample of the nuclear and cytoplasmic transcript half-lives for the 3'UTR of the MAFG gene. The half-life distributions are given at the top and right hand side.

3'UTRs with the same strand orientation, we obtained 61,834 3'UTRs. For robustness, we consider only 8119 of these 3'UTRs with an average number of at least 30 reads per time point in each cellular compartment of the time series experiment. The 3'UTR read counts are highly correlated between time points of both the nuclear and cytosolic fractions, indicating that 4sU labeling does not generally perturb gene expression patterns (Fig A in S2 Appendix). We noticed that the read distribution within a 3'UTR was often multimodal, indicating the existence of alternative 3'UTR isoforms. In addition, we found many reads clustering inside gene bodies, likely due to A-rich sequences acting as internal priming sites for the oligo-dT primer

utilized for the sequencing library preparation. We therefore performed a second, refined analysis in which we defined densely covered, highly confined read clusters ("peaks") along the entire genome (Section 4 and Figs F-G in S1 Appendix). We identified 98,102 distinct peaks within 3'UTRs, of which we selected 10,150 with robust expression, i.e., at least 30 average counts in each cellular compartment and time-series experiment. Moreover, 1,262,263 peaks were found in non-3'UTRs, of which 1651 were robustly expressed according to the same criterion. Of these 1651 peaks, 1057 are located within an annotated gene region (334 introns, 174 exons and 549 either overlapping with both an exon and intron, or with regions that have no transcript annotation). Subsequently, modeling was performed for 3'UTRs and peaks separately.

## A two-compartment model of RNA metabolism

We model the life cycle of a mature (polyadenylated) transcript by four metabolic parameters. First, RNA is synthesized in the nucleus at a rate $\mu$. Then, the mature transcript is either exported to the cytosolic compartment at a rate $\tau$ or eventually degraded in the nucleus at a nuclear degradation rate $v$ (Fig 1B). By "export" we mean all processes that take place after polyadenylation until passage through the nuclear pore complex. Exported transcripts are finally degraded at rate $\lambda$ irrespective of the function they fulfill in the cytosol (Fig 1B). Our model assumes that these four rates are constant over time and unaffected by 4sU labeling.

Denote by $N = N(t)$ and $C = C(t)$ the time-dependent, cell-averaged nuclear respectively cytosolic RNA abundances of a given RNA population. We will describe their dynamics by a two-compartment ordinary differential equation system (Fig 1B):

$$\frac{dN}{dt} = \mu - (v + \tau)N \tag{1}$$

$$\frac{dC}{dt} = \tau N - \lambda C \tag{2}$$

The system has a closed-form solution for any choice of initial conditions at $t = 0$. In steady state, the nuclear and cytosolic RNA abundances are given by $N_\infty = \frac{\mu}{v+\tau}$ and $C_\infty = \frac{\tau}{\lambda} \cdot N_\infty$. Unlike other RNA labeling methods, SLAM-seq does not require the separation of labeled and unlabeled RNA fractions. As a consequence, SLAM-seq provides, for each transcript, more accurate measurements of the newly synthesized RNA by total RNA ($\frac{new}{total}$) ratio in the nucleus, $n(t)$, respectively the cytosol, $c(t)$. These two ratios can be derived from Eqs (1) and (2) (Fig 1C, Methods):

$$n(t) = 1 - e^{-(v+\tau)t} \tag{3}$$

$$c(t) = 1 - \frac{\lambda e^{-(v+\tau)t} - (v+\tau)e^{-\lambda t}}{\lambda - (v+\tau)} \tag{4}$$

Note that Eq (3) involves only one parameter, namely the sum of the export rate and the nuclear degradation rate, $v + \tau$. We will henceforth call this quantity nuclear removal rate. Moreover, the quotient in Eq (4) is determined by two parameters, the nuclear removal rate and the cytosolic degradation rate $\lambda$. Eqs (3) and (4) are the building blocks of our analysis, because they allow us to fit the dynamics of individual RNA populations with only two parameters.

## Estimation of new/total RNA ratios

The ratios $n(t)$ and $c(t)$ required for parameter estimation must be derived from the labeling pattern of all reads mapping to a certain 3'UTR or peak. However, estimating these ratios is challenging due to the low 4sU incorporation efficiency, which is reported to be in the range of 2% [29]. As a result, some reads from new transcripts will not show a single labeling-induced nucleotide conversion and will appear as pre-existing. Thus, a naive count of labeled reads in a SLAM-seq RNA library underestimates the relative abundance of new transcripts. In addition, sequencing errors can mimic or mask converted 4sU positions.

To accurately estimate the labeling efficiency, we first excluded all known SNP positions and positions enriched for post-transcriptional RNA-editing. As in previous approaches [26, 35], we use a binomial mixture model to robustly estimate the labeling efficiency in a given sample by an Expectation-Maximization (EM) algorithm [36] (Section 5 in S1 Appendix). We find that the labeling efficiency rises from about 4.3% (nucleus) and 3.2% (cytosol) at 15 min to 7.3% (nucleus) and 6.7% (cytosol) at 180 min (Fig H in S1 Appendix).

Note that current models of the RNA labeling process—including ours—assume that all new RNA molecules present in a sample are labeled with the same efficiency [28, 35, 37]. As labeling efficiency necessarily increases upon addition of 4sU, we investigated the potential bias introduced by this assumption. We found that this bias must not be ignored for labeling periods shorter than 10 min, while the estimates in our experimental setup are not affected substantially (Section 6 and Figs I-K in S1 Appendix). In addition to the labeling efficiencies, the EM algorithm also estimates the $\frac{new}{total}$ RNA ratios for each 3'UTR. These serve as observations to which we fit the parameters of the two-compartment model.

## Estimation of metabolic parameters

Using Eqs (3) and (4), we fitted the nuclear removal rate $\nu + \tau$ and the cytosolic degradation rate $\lambda$ to the estimated $\frac{new}{total}$ RNA ratios (Fig 1C). We found that a variance-stabilizing transformation of the $\frac{new}{total}$ ratios prior to fitting improved the robustness and precision of the estimates (Section 7 and Fig L in S1 Appendix). We verified the goodness of fit by inspecting the homogeneity and independence of the residuals (Fig 1D). Error bounds were constructed by Markov Chain Monte Carlo sampling (Fig 1E). For ease of interpretation, we report the nuclear removal rate $\nu + \tau$ and the cytosolic degradation rate $\lambda$ as their corresponding nuclear half-life, $\frac{\ln 2}{\nu + \tau}$, and cytosolic half-life, $\frac{\ln 2}{\lambda}$, respectively. The half-life estimates of the two replicate time-series are highly correlated (Spearman's $\rho$: 0.98 for nuclear and 0.70 for cytosolic half-life, Section 1 and Figs C-D in S2 Appendix). Subsequently, rate estimates obtained from the two replicates were averaged.

We define stringent quality criteria for our metabolic rate estimates in terms of constant expression, goodness of fit, width of the error bounds, and agreement between replicates (Methods). Of the 8,119 3'UTRs, we found 1,297 whose nuclear removal rate could be reliably estimated (Fig 2A and E in S2 Appendix, S1 Table). Of these, 251 3'UTRs also had reliable estimates for the cytosolic degradation rate (Fig 2A and E in S2 Appendix, S2 Table). We report all results of our analyses for reliable 3'UTRs and the corresponding numbers for all quantified 3'UTRs are listed in parentheses. Applying the same procedure to peak regions, we obtained in 261 peaks with reliable nuclear removal rates and 51 with reliable cytosolic degradation rates.

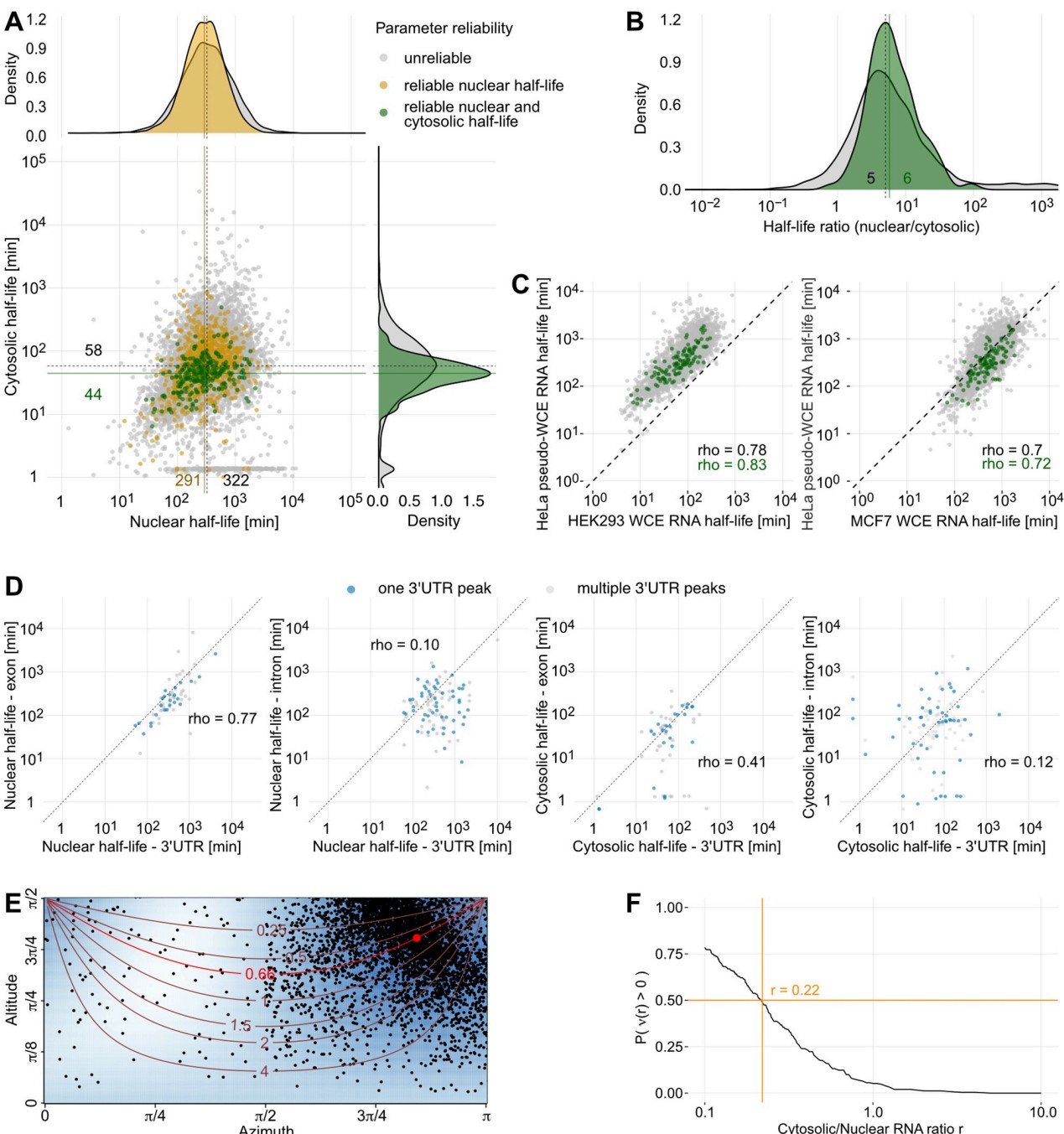

**Fig 2. Nuclear RNA half-lives are longer than cytosolic half-lives.** (A) Nuclear and cytosolic RNA half-life estimates of 3'UTRs. Half-lives were averaged over both measured time series. Gray dots represent unreliable estimates, yellow dots correspond reliable estimates for nuclear half-life, and green dots portray reliable estimates for both nuclear and cytosolic compartment (see Methods for reliability criteria). The half-life estimate distributions are given at the top and right hand side, color representation as in scatterplot. The solid and dashed lines indicate median half-life estimates of reliable and all 3'UTRs, respectively. (B) Nuclear by cytosolic half-life ratios of all 3'UTRs (gray) and 3'UTRs with reliable half-life estimates (green) for both compartments. The solid and dashed lines indicate median half-life ratios of reliable and all 3'UTRs, respectively. (C) Comparison between half-life estimates by our two-compartment model and whole-cell extract half-life measurements from Schueler et al. (2014) [38]. Our nuclear and cytosolic RNA estimates were summed to generate pseudo-whole-cell estimates. Gray dots represent unreliable estimates, green points represent reliable estimates. (D) Correlation of half-lives between 3'UTRs and exonic or intronic peaks of the same gene. Only 3'UTRs and peaks that had one unique gene annotation were considered. Data points are colored by whether the 3'UTR harbors one (gray) or multiple (blue) expressed 3'UTR peaks. (E) Estimation of the $\frac{cyt}{nuc}$ RNA ratio using a spherical median. Each dot represents the angular coordinates of a 3d unit vector, which in turn is determined as the normal vector of a plane spanned by three triplets $(t_{g_i}, -n_{g_i}, -c_{g_i})$ of three randomly sampled 3'UTRs $g_i$, $i = 1, 2, 3$.

Contour lines show locations of constant $\frac{cyt}{nuc}$ RNA ratio. The red dot is the spherical median of the sampled normal vectors and corresponds to a ratio of 0.66. Azimuth angles $\psi \in [\pi, 2\pi]$ correspond to normal vectors $v$ in which not all entries are positive and are omitted. (F) Fraction of non-negative nuclear degradation rate estimates as a function of the $\frac{cyt}{nuc}$ RNA ratio. Assuming at least 50% positive estimates (orange horizontal line) leads to a maximum admissible ratio of 0.2 (orange vertical line).

## Nuclear half-life is substantially longer than cytosolic half-life

We compared the reliable half-life estimates of the nuclear and cytosolic compartments (Fig 2A and E in S2 Appendix). Both distributions are right-skewed, with a median of 291 min in the nucleus (interquartile range, IQR: 171–481 min) and a median of 44 min in the cytosol (IQR: 30–62 min). These distributions look similar when considering all 3'UTRs, with a median of 322 min in the nucleus (interquartile range, IQR: 171–481 min) and a median of 58 min in the cytosol (IQR: 28–117 min). Furthermore, the median ratio of the reliable nuclear versus cytosolic half-lives of a 3'UTR is 5.76 (Fig 2B and F in S2 Appendix). These findings imply that nuclear export is much slower than cytosolic degradation.

Next, we compared our half-lives with estimates from Schueler et al. (2014) [38], which are based on whole-cell extracts (WCE) from MCF7 and HEK293 cells measured by 4sU labeling and biotinylation (Fig 2C, Section 2 and Fig G in S2 Appendix). The sum of nuclear and cytosolic half-lives correlates highly with the WCE half-lives measured in HEK293 (reliable 3'UTRs: Spearman's $\rho$=0.83, all 3'UTRs: Spearman's $\rho$=0.78) and MCF7 (reliable 3'UTRs: Spearman's $\rho$=0.72, all 3'UTRs: Spearman's $\rho$=0.7) cells. Notably, the estimates from HEK293 cells are systematically shorter than our reliable HeLa-S3 half-lives by a median factor of 0.31 (all 3'UTRs: 0.21), while the opposite is the case for the MCF7 cells with a median factor of 1.73 (all 3'UTRs: 1.18). The systemic differences may be due to actual differences in RNA metabolism between the different cell lines or due to differences in scaling factors between the experiments. For further validation, we also compared our half-lives with estimates from Wu et al. (2019) [39], which were derived from WCE SLAM-seq data of K562 cells (Section 2 and Fig H in S2 Appendix). The half-lives correlate well (reliable 3'UTRs: Spearman's $\rho$=0.7, all 3'UTRs: Spearman's $\rho$=0.65) but without a systematic shift.

We expect exons to share their metabolism with the 3'UTRs of the same transcript. Reassuringly, the metabolic rates of peaks located in exonic regions correlate well with their 3'UTRs (Fig 2D). In contrast, metabolic rates of peaks located in intronic regions show only weak or no correlation with their corresponding 3'UTRs (Fig 2D), as most introns are spliced out co-transcriptionally. We hypothesize that intronic peaks detected in the cytosol arise from intron retention events, unknown alternative exons or unknown non-coding RNA species.

## mRNA is more abundant in the nucleus than in the cytosol

The systematic difference in nuclear and cytosolic half-lives has consequences on the subcellular distribution of mRNA. According to our two-compartment model (Methods), the ratio of steady-state cytosolic versus nuclear RNA abundance levels is $\frac{C_\infty}{N_\infty} = \frac{\tau}{\lambda}$. This $\frac{cyt}{nuc}$ RNA abundance ratio is at most as large as the quotient of the cytosolic and the nuclear half-life, $\frac{\tau+v}{\lambda}$. The median of this ratio $\frac{\tau+v}{\lambda}$ across all reliable transcripts is 0.17 (median of its inverse $\frac{\lambda}{\tau+v} = 5.76 \approx 6$, Fig 2B). We pursued three independent approaches to verify this finding (Section 3 in S2 Appendix). First, we exploit that the relative abundances $t_g$ of a transcript $g$ in the whole-cell RNA are the sum of the corresponding relative abundances $n_g$ and $c_g$ in the nuclear and cytosolic fractions when scaling them to absolute molecule numbers by transcript-independent factors $T$, $N$ and $C$, $Tt_g = Nn_g + Cc_g$. Since there are many outlier triples, we performed a robust non-

parametric linear regression (Fig 2E, Section 3 in S2 Appendix). According to this method, the $\frac{cyt}{nuc}$ RNA ratio is 0.66.

Second, we computed the nuclear degradation rate estimates as a function of the $\frac{cyt}{nuc}$ RNA ratio for all 3'UTRs with reliable half-life estimates (Fig 2F, Section 3 in S2 Appendix). As nuclear degradation rates are non-negative, the majority of their estimates should also be positive. If we require merely 50% of the degradation rate estimates to be positive, the smallest admissible $\frac{cyt}{nuc}$ RNA ratio is 0.22.

Third, we normalized our SLAM-seq time series samples to spike-in counts (Section 3 in S2 Appendix). The resulting $\frac{cyt}{nuc}$ RNA ratios ranged from 0.28 to 0.7 for the individual samples. In summary, all methods indicate that the amount of nuclear mRNA exceeds the cytosolic one by a factor of at least 1.4.

## 3'UTR length is a major determinant of mRNA half-life

We investigated several gene-specific features for their association with transcript half-life (Section 4 in S2 Appendix). Our reliable nuclear and cytosolic transcript half-life estimates are neither significantly correlated with GC content nor transcript length (Figs J-K in S2 Appendix). In contrast, the RNA half-lives of both compartments showed mild but significant positive correlations with exon count (reliable 3'UTRs nucleus: Spearman's $\rho$=0.24, reliable 3'UTRs cytosol: Spearman's $\rho$=0.3, Fig L in S2 Appendix). We also observed an anticorrelation of 3'UTR length with nuclear RNA half-life (reliable 3'UTRs: $\rho$=-0.36, all 3'UTRs: $\rho$=-0.26, Fig 3A and M in S2 Appendix) and CDS length with nuclear RNA half-life (reliable 3'UTRs: $\rho$=-0.16 all 3'UTRs: $\rho$=-0.22, Fig 3A and N in S2 Appendix). A similar association between whole-cell RNA half-lives and 3'UTR lengths in yeast was observed by Cheng et al. (2017) [40]. Note that we did not observe a similar association between 3'UTR length and cytosolic RNA half-life (reliable 3'UTRs: Spearman's $\rho$=0.03 and insignificant, all 3'UTRs: Spearman's $\rho$=-0.04, Fig 3A and M in S2 Appendix), as well as CDS length and cytosolic RNA half-life (reliable 3'UTRs: Spearman's $\rho$=0.08 and insignificant, all 3'UTRs: Spearman's $\rho$=-0.02, Fig 3A and N in S2 Appendix).

It has been hypothesized that the length of the 3'UTR has a significant effect on the half-life of nuclear RNA [40]. To test this, we examined the half-lives of different 3'UTR isoforms of the same transcript (Section 5 in S2 Appendix). Since these isoforms share many sequence features and other confounding factors (e.g. epigenetic), comparing their half-lives should reveal the effect of 3'UTR length with high precision. We considered peaks located in the same 3'UTR but separated by at least 100 nucleotides to be different 3'UTR isoforms. We identified 118 3'UTRs with more than one quantifiable isoform. Of those, the longer 3'UTR isoforms generally have shorter nuclear half-lives (Fig 3B), and this difference cannot be explained by technical variation (as observed between the two replicate time-series measurements, Figs O-P in S2 Appendix). In agreement with our previous observation (Fig 3A), 3'UTR isoform length had only a minor effect on cytosolic RNA half-life (Fig 3B).

## RBP-bound transcripts differ in nuclear turnover rates

RNA binding proteins (RBPs) are involved in many regulatory processes of RNA metabolism. For instance, they maintain stability, regulate splicing, and control the localization of bound RNA molecules [41–43]. We retrieved eCLIP data for 120 RBPs deposited on ENCODE [44, 45] and defined confidently RBP-bound and RBP-unbound transcripts (Fig 3C, Section 6 in S2 Appendix). For RNAs with reliable half-life estimates, we then tested whether there was a difference between the RBP-bound and unbound groups. In total, 20 RBPs showed significant effects on nuclear RNA half-lives (Fig Q in S2 Appendix). Unexpectedly, only one protein,

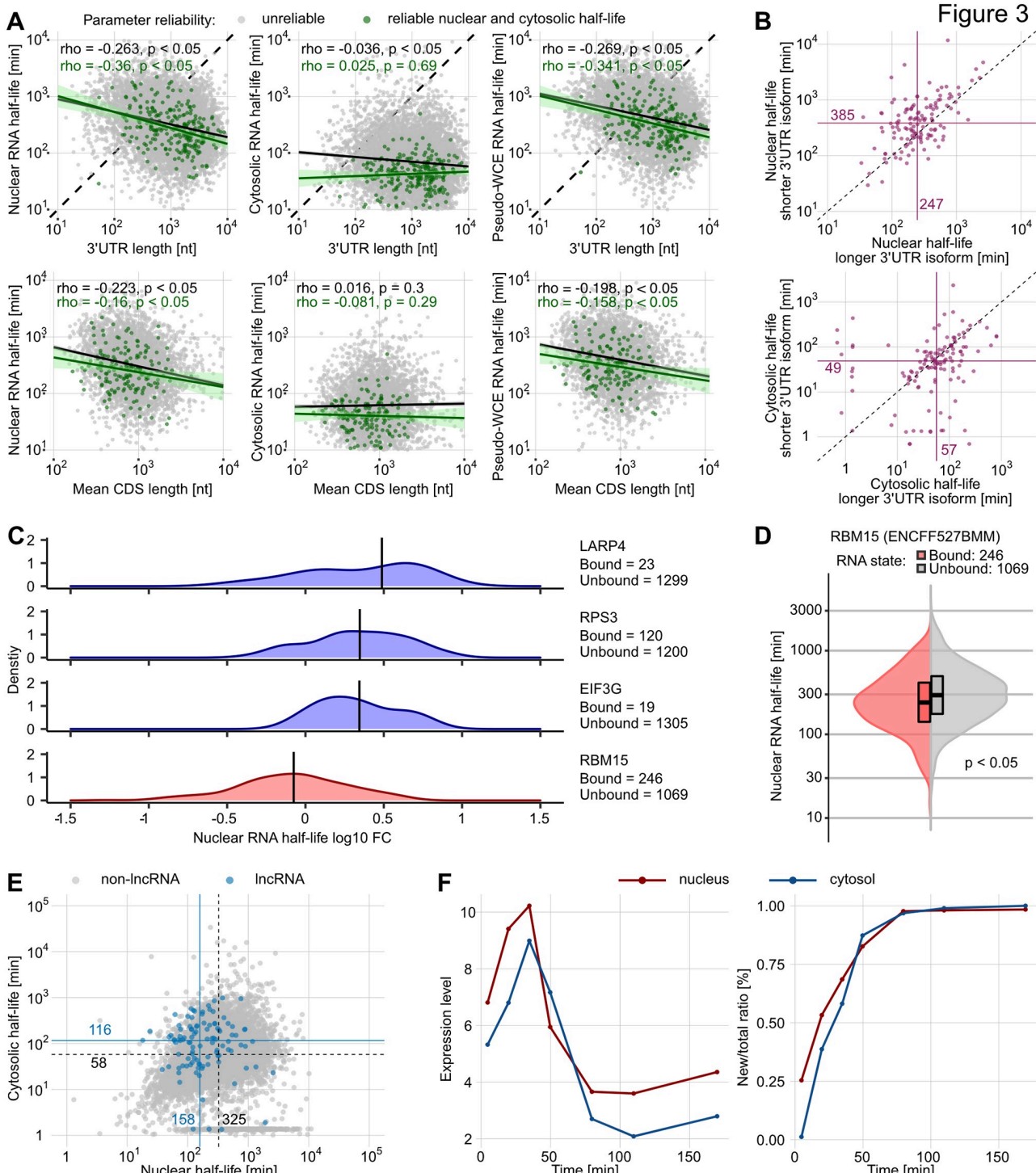

**Fig 3. Factors correlating with nuclear and cytosolic half-lives.** (A) Correlation of half-life estimates with gene-specific parameters. Shown are the nuclear and cytosolic half-lives plotted against the 3'UTR lengths and the CDS lengths per gene. All 3'UTRs are shown in gray and 3'UTRs with reliable half-life estimates are shown in green with corresponding regression lines. (B) Comparison between RNA half-lives of short and long isoforms of a 3'UTR region (242 peaks from 118 3'UTRs). The medians are indicated by the violet lines with corresponding text labels. (C) Nuclear half-lives differ between different RBP-bound transcripts. Density plots show the distribution of nuclear half-life fold-changes (FC) relative to the global median (all bound and unbound transcript half-lives) for selected RBP-bound transcripts (eCLIP data retrieved from ENCODE). Distribution densities with median fold-change above or below the global median are colored in blue and red, respectively. Black bars indicate the median of the respective distribution. The number of bound and unbound transcripts is given for each RBP. (D) Violin plot of the nuclear half-life distributions of transcripts

that are either bound (red) or unbound (gray) by RBM15 (p < 0.05, Wilcoxon test; eCLIP data retrieved from ENCODE). (E) Nuclear and cytosolic RNA half-life estimates of lncRNAs. Blue dots represent lncRNA estimates and gray dots portray all other 3'UTRs. The blue lines indicate the median half-lives of the lncRNAs, the black lines indicate the median half-lives of all quantified 3'UTRs, respectively. (F) Expression level and $\frac{new}{total}$ RNA ratios of the 'supernova' gene SGK1.

RBM15, appeared to promote nuclear RNA export. RBM15-bound transcripts have a shorter median nuclear RNA half-life than the unbound transcripts (bound: 206 min, unbound: 315 min, Fig 3C and 3D). Previous evidence supports our findings as RBM15 has been associated with NXF1-dependent nuclear export [46, 47]. The putative binding of all other RBPs was correlated with longer nuclear half-lives. Among them, LARP4, EIF3G and RPS3 had the most retarding effect on export from the nucleus (Fig 3C and Q in S2 Appendix). Further, we found that the putative binding of only 2 RBPs, U2AF1 and DDX24, had a prolonging effect in the cytosol (Fig R in S2 Appendix).

## Long non-coding RNAs have a higher cytoplasmic stability than mRNA

Besides mRNA, we also detected 106 polyadenylated transcripts annotated as long non-coding RNAs for which we could estimate metabolic rates (Section 7 in S2 Appendix). Interestingly, the detected lncRNAs exhibit a shorter median nuclear half-life than mRNAs (lncRNA: 158 min vs all other 3'UTRs: 325 min, Fig 3E and S in S2 Appendix). Conversely, the median cytosolic half-life of lncRNAs is, compared to mRNAs, substantially longer (lncRNA: 116 min vs all other 3'UTRs: 58 min, Fig 3E and S in S2 Appendix). Since many lncRNAs perform their function in the cytosol, such as maintaining mRNA stability and modulating transcript translation (reviewed in [48]), longer cytoplasmic half-lives might reflect these roles.

## Nuclear export of stress-response genes is rapid

While the expression of most genes is not affected by 4sU labeling, we identified a collection of 27 transcripts that are strongly induced immediately after the labeling onset and silenced shortly afterwards (Fig 3F, S3 Table). According to Uniprot [49], 14 of these genes are linked to inflammation, cell survival, apoptosis and growth. Note that our two-compartment model assumes steady-state conditions and, therefore, cannot be applied to this group of genes. However, the time interval between maximum and the subsequent minimum expression levels is an upper limit for the RNA half-life in the respective compartment. In the nucleus, this interval is approximately 60 min, which is substantially shorter than our reported median nuclear mRNA half-life of 322 min (Fig 2A). As all these labeling-induced transcripts also show a similar, fast relaxation pattern back to the initial expression level, we hypothesize that these transcripts are exported by an alternative nuclear export mechanism as reported previously [50].

## The two-compartment model is robust

Given the surprisingly long nuclear half-lives, we thoroughly investigated potential sources of bias, namely the accuracy of subcellular RNA fractionation and the neglect of nuclear retention. Because the endoplasmic reticulum (ER) is physically attached to the nucleus, it could be co-purified with the nucleus upon subcellular fractionation. Since the rough ER is a site of cytosolic mRNA translation, such a co-purification would enrich pre-existing cytosolic transcripts in the nuclear fraction. A consequence would be an unintended reduction of the nuclear $\frac{new}{total}$ RNA ratios, leading to an upward bias of nuclear half-life estimates. Conversely, cytosolic $\frac{new}{total}$ RNA ratios would be increased, biasing cytosolic half-lives downward. We examined the metabolic rates of 136 genes annotated as part of an ER structure according to gene

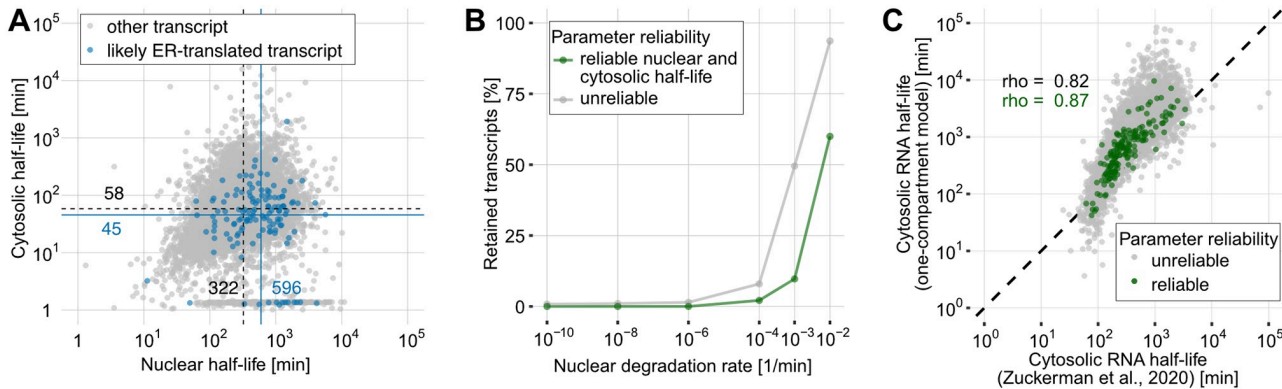

**Fig 4. Potential biases of the two-compartment model estimates.** (A) Nuclear and cytosolic RNA half-life estimates of ER-translated transcripts according to gene ontology annotation. Blue dots represent estimates for ER-translated transcripts and gray dots portray all other 3'UTRs. The blue and black lines with corresponding text labels indicate the median half-lives of ER-translated transcripts and all other 3'UTRs, respectively. (B) Line plot showing the percentage of retained transcripts for a range of postulated nuclear degradation rates. A transcript was defined as retained if its nuclear retained fraction was at least 5%. (C) Correlation of a simple exponential cytosolic decay model fit results with cytosolic half-lives obtained through SLAM-seq on the cytosolic fraction of MCF7 cells [47]. Gray dots represent expressed 3'UTRs and green dots portray 3'UTRs that passed our reliability criteria for the cytosolic compartment.

ontology. The median nuclear half-life for these genes was 596 minutes (compared to the global median of 322 min), and the median cytosolic half-life was 45 minutes (compared to the global median of 58 min; Fig 4A and T in S2 Appendix). For this effect to be relevant, the majority of the cytosolic transcripts would have to be purified together with the nuclear RNA. However, we cannot completely rule out this possibility.

Next, we examined whether a partial nuclear retention of transcripts might affect our half-life estimates. Our two-compartment model assumes that the nuclear export of an RNA population can be appropriately described by a single, constant export rate $\tau$. But it seems plausible that a certain fraction of these transcripts are exported at a very low efficiency different from $\tau$, be it due to the lack of export factors, because of erroneous splicing, the use of alternative 3' ends, or as a mechanism to buffer against transcriptional bursts [1]. Therefore, we challenged our model by introducing an additional retention rate $r$, which specifies a fraction of transcripts (of a given RNA population) that are retained in the nucleus, e.g., this fraction has an export rate of 0 (Section 8 in S2 Appendix). To prevent an infinite accumulation of nuclear RNA, we also included a non-zero nuclear degradation rate $v$ which we varied extensively. Parameter fitting was performed using MCMC (Section 8 in S2 Appendix). For the vast majority of genes, the estimated percentages of retained mRNA are negligible assuming that the nuclear mRNA degradation rate is substantially lower than the nuclear export rate (Fig 4B). This observation indicates that RNA retention is not a major cause for low nuclear export rate estimates.

## The time-scale of RNA metabolism is mainly determined by nuclear processes

So far, RNA half-lives have been estimated from either WCE or cytosolic extracts using an exponential decay model [22, 26, 35]. The biological features that correlate with the half-lives obtained by such models were mainly attributed to cytosolic RNA degradation. However, we have shown that slow nuclear export contributes to a major part of RNA lifespan (Fig 2A). As the half-lives estimated from whole-cell extracts are essentially the sum of nuclear and cytosolic RNA half-life estimates (Fig 2C), we hypothesized that fitting a one-compartment model to

either whole-cell extracts or the cytosolic RNA fractions would yield estimates that mainly reflect the nuclear half-life (Section 9 and Fig U in S2 Appendix). In fact, half-life estimates obtained by fitting a simple exponential decay model to our cytosolic data show a higher correlation with nuclear half-lives (reliable 3'UTRs: Spearman's $\rho$=0.93, all 3'UTRs: Spearman's $\rho$=0.84, Fig U in S2 Appendix) than cytosolic half-lives (reliable 3'UTRs: Spearman's $\rho$=0.61, all 3'UTRs: Spearman's $\rho$=0.65, Fig U in S2 Appendix) obtained from our two-compartment model. Furthermore, compared to the cytosolic half-lives obtained from reliable 3'UTRs by our two-compartment model, the cytosolic half-life estimates of the one compartment model are biased upward by a median factor of 15 (Fig U in S2 Appendix). Notably, the half-life estimates obtained by fitting a simple exponential decay model to our cytosolic data correlate best with the sum of our nuclear and cytosolic half-life estimates (reliable 3'UTRs: Spearman's $\rho$=0.97, all 3'UTRs: Spearman's $\rho$=0.95, Fig U in S2 Appendix).

Our results were further supported by the analysis of a previously published SLAM-seq dataset obtained from MCF7 cells by Zuckerman et al. (2020) [47] (Section 2 in S2 Appendix). In this study, RNA half-lives were estimated from the cytosolic RNA fraction only. In consistency with our previous findings, we observed a high agreement with our nuclear half-life estimates (reliable 3'UTRs: Spearman's $\rho$=0.88, all 3'UTRs: Spearman's $\rho$=0.84, Fig I in S2 Appendix). The agreement was even higher when compared to the sum of our nuclear and cytosolic half-life estimates (reliable 3'UTRs: Spearman's $\rho$=0.89, all 3'UTRs: Spearman's $\rho$=0.86, Fig I in S2 Appendix). At the same time, the correlation with our cytosolic half-life estimates was substantially lower (reliable 3'UTRs: Spearman's $\rho$=0.45, all 3'UTRs: Spearman's $\rho$=0.42; Fig I in S2 Appendix). Since our one-compartment cytosolic half-life estimates agree well with the data by Zuckerman et al. (2020) [47] (reliable 3'UTRs: Spearman's $\rho$=0.87, all 3'UTRs: Spearman's $\rho$=0.82, Fig 4C and V in S2 Appendix), this implies that proper subcellular RNA fractionation and two-compartment modeling are necessary for adequate quantification of RNA metabolism.

## Discussion

In this work, we quantified nuclear and cytosolic RNA metabolism on a genome-wide scale by applying a two-compartment model to SLAM-seq time-series data. Although conceptually simple, the actual implementation of such a model is challenging, as the reliability of the metabolic parameter estimates critically hinges on careful preprocessing, removal of potential biases, robust parameter estimation and quality filtering. We obtained high quality nuclear half-life estimates (median half-life of reliable 3'UTRs: 291 min, all 3'UTRs: 322 min) and cytosolic half-life estimates (median half-life of reliable 3'UTRs: 44 min, all 3'UTRs: 58 min) for 8,119 mRNAs. Our pseudo-WCE RNA half-life estimates correlate highly with estimates from three different studies [38, 39, 47]. This suggests that our two-compartment model is consistent with previous one-compartment models, but can resolve the nuclear and cytosolic contribution to RNA metabolism. Interestingly, we find systematic shifts in half-lives between cell lines, which could either be explained by different cell physiology (e.g., compartment size) or technical biases specific to the quantification protocols of labeled RNA.

Our main finding is that an mRNA molecule generally spends most of its lifespan in the nucleus and not in the cytosol. We have shown that most of a cell's mRNA is, therefore, localized in the nucleus. Our results contrast with the much shorter transit times through the nuclear pore complex observed in smFISH experiments (on the order of seconds) [17]. However, these results are not contradictory because the time span we define as the nuclear half-life includes not only the translocation time through the NPC, but also the time required for dissociation from chromatin and diffusion/transport to the NPC. Thus, the long nuclear half-lives

suggest that translocation through the NPC is not the bottleneck for RNA export. Slow nuclear export has been identified as a mechanism to buffer cytoplasmic mRNA levels against transcriptional bursts [1]. Since our nuclear half-life estimates are considerably longer than the commonly assumed timescales of transcriptional bursts for most transcripts, this suggests a general buffering function of nuclear export.

The speed of nuclear RNA export also has functional consequences for gene regulation. The absolute rate at which a cell changes its transcript or protein abundance is essentially determined by the slowest step in the life of that molecule. Genes involved in fast cellular processes have been shown to have shorter cellular half-lives on average than, for example, housekeeping genes [28]. However, some rapid adaptations must occur on a much shorter time scale than the typical nuclear mRNA half-lives. Therefore, a rapid regulatory response must either act at the protein level rather than on the RNA level, or there must be very rapid, transcript-specific alternative export mechanisms for this purpose. Indeed, alternative transport molecules and pathways for stress response genes are being discussed [50].

Although 4sU labeling is considered minimally perturbing (as also confirmed in our case), it is known to elicit some stress responses [35, 51, 52]. Consistent with this, we have discovered 27 genes that show a rapid increase in transcript levels in both the nucleus and cytosol almost immediately after the onset of 4sU labeling, followed by a similarly rapid decrease to the original levels. To elucidate alternative export mechanisms, it will be promising to investigate the RNP composition of fast-responding transcripts under normal and stress conditions.

Analysis of eCLIP RNA-protein interactions revealed RBPs that appear to modulate nuclear half-life upon binding. RBM15 was the only RBP whose bound transcripts were exported faster to the cytosol and had correspondingly shorter nuclear half-lives. Indeed, RBM15 has previously been implicated in the NXF1 export pathway [46]. However, its exact function remains elusive. Surprisingly, all other RBPs with a significant effect appeared to prolong RNA export. We speculate that the targets of these RBPs are bound while still chromatin-associated. Nevertheless, the eCLIP data must be interpreted with caution, as they were obtained from whole cell extracts and different cell lines. The subcellular localization of some factors such as LARP4 is cytosolic according to Uniprot [49]. Thus, their association with nuclear half-life is unlikely to be causal. Looking for other determinants of RNA half-life, we found a weak negative correlation between 3'UTR length and RNA half-life, consistent with previous findings in S. pombe [40]. Our data allowed an in-depth study of this effect. It contains 118 genes with at least two different 3'UTR isoforms whose metabolism could be reliably quantified. While there were no substantial differences in cytosolic half-lives, we found that the median nuclear RNA half-life for the shorter 3'UTR was 1.5 times longer than the half-life of its longer counterpart. The role of 3'UTR length in this context is controversial [53]. While some studies report a negative association between 3'UTR length and half-life [31, 54], others did not observe such a trend [53]. Our comparison of 3'UTR transcript isoforms provides additional evidence for the former.

In this work, we have developed a two-compartment model that captures key aspects of RNA metabolism. We anticipate that our method will elucidate the functions of RBPs in mRNA export by comparing RNA metabolism in control and functional knockouts of a putative export factor. It will also allow the dissection of different export pathways and their transcript specificity. However, the simplifying assumptions of the two-compartment model limit its scope. For example, we have neglected the breakdown of the nuclear envelope during mitosis. During this phase, it is possible that nuclear and cytosolic RNA fractions mix by diffusion. Therefore, an extension of the model with a cell cycle phase-specific diffusion parameter is promising. Complementary to this, SLAM-seq data should be acquired from a synchronized cell population. Another limitation of our current model is the steady-state assumption, which

will require extension to non-stationary conditions in future work. In conclusion, we have shown that nuclear export is a major contributor to RNA lifespan. To our knowledge, all previous studies have explained transcript half-life by cytosolic degradation. Therefore, future studies should consider the compartmentalization of a eukaryotic cell as a major factor in RNA metabolism.

## Material and methods

### Cell culture and subcellular fractionation

Hela-S3 cells were purchased from ATCC and cultured in DMEM supplemented with 10% heat inactivated FBS without antibiotics and incubated at 37˚C and 5% CO2 Cells were tested regularly for mycoplasma with the PlasmoTest kit (InvivoGen) and are mycoplasma negative. We labeled newly synthesized RNA of asynchronously proliferating cells with 4-thiouracil (4sU; Carbosynth) at a concentration of 500 μM. Apart from one report on 4sU labeling triggering a ribosomal stress response in human cells [51], 4sU labeling at these concentrations is generally considered minimally perturbing [24, 29, 55]. 45 million cells were used per sample. After 0, 15, 30, 45, 60, 90, 120 or 180 min of 4sU labeling, we performed subcellular fractionation into nuclear and cytoplasmic fractions as described in Nojima et al. (2016) [56] with modifications as follows: the cells were rinsed on the cell culture plates twice with 20 ml of ice-cold DPBS twice and scraped in 10 ml ice-cold DPBS and transfered into a 15-ml tube. The cells were pelleted at 400 x g or 5 min at 4˚C resuspended in 4 ml of ice-cold HLB+N buffer (10mM Tris-HCl (pH 7.5), 10 mM NaCl, 2.5 mM MgCl2 and 0.5% (vol/vol) NP-40) followed by an incubation on ice for 5 min. The cell pellet was underlaid with 1 ml of ice-cold HLB+NS buffer (10 mM Tris-HCl (pH 7.5), 10 mM NaCl, 2.5 mM MgCl2, 0.5% (vol/vol) NP-40 and 10% (wt/vol) sucrose). The nuclei were pelleted by centrifugation at 400 x g for 5 min at 4˚C. The supernatant was the collected and lysed with 3 volumes of Trizol LS (Life technologies) for cytoplasmic fraction. The nuclei were washed with 15 ml ice-cold DPBS and lysed in Qiazol (Qiagen). For the total fractions we lysed the metabolically labeled cells directly in Qiazol (Qiagen). To rule out cross-contamination between the fractions, the purity of the fractions was assessed by Western Blots. GAPDH and $\alpha$-tubulin were used as markers for the cytosolic fraction and RNA Pol II CTD—Ser2P, U1snRNAP70 and histone H3 as markers for the nuclear fraction (Fig B in S1 Appendix).

### RNA preparation and sequencing

Synthetic spike-ins were produced and purified as described previously [57, 58]. The preparation protocol was modified to include in vitro poly(A) tailing with E. coli Poly(A) Polymerase (NEB) to make the spike-ins compatible with the NGS library preparation. We added equal amounts of synthetic spike-ins to all the RNA fraction lysates prior to RNA isolation. RNA was isolated according to the manufacturer's protocol and then dissolved in nuclease-free water with 1 mM DTT. The carboxyamidomethylation reactions of 4sU were set up as described by Herzog et al. [29]. Briefly described here, we set up the reaction with 5 μg RNA in nuclease-free water with1 mM DTT; 50% DMSO (Sigma); 50 mM sodium phosphate buffer pH 8.0; and 10 mM iodoacetamide (Pierce). The reactions were incubated at 50˚C for 15 min and then quenched by 100 mM DTT. The conversion efficiency of 4sU was evaluated by absorption spectra before and after treatment with IAA (Fig A in S1 Appendix).

The RNA was then precipitated with ethanol to clean up and concentrate. Next generation sequencing (NGS) libraries were prepared from 500 ng RNA with the QuantSeq 3' mRNA-Seq Library Prep Kit REV for Illumina (Lexogen) according to the manufacturer's instructions, but with two modifications. In brief, the reverse transcription reaction temperature was

increased to 50˚C and the incubation was extended to 60 min. The NGS libraries were pooled and sequenced as single-read with the custom sequencing primer provided with the QuantSeq NGS library preparation kit with 100 cycles on a HiSeq2000 sequencer (Illumina).

## Sequencing data pre-processing

We chose SlamDunk [34] for read mapping, after comparison with results from bowtie2 [59] and hisat2 [60] (Fig C in S1 Appendix). Reads were mapped to human reference genome hg19 with SlamDunk (version 0.3.0, settings: -n 100, -m; see Section 2 and Figs C and D in S1 Appendix for details). Count statistics for each 3'UTR were obtained by aggregating all mapped reads overlapping with a given 3'UTR. To prevent double counting, overlapping 3'UTRs on the same DNA strand were merged.

Most genes have multiple poly-adenylation sites, which can give rise to multiple 3'UTR isoforms [61]. Moreover, the annotation of 3'UTRs might be incomplete, and there might be additional pA sites in the genome arising from non-coding and cryptic transcripts would have been overlooked by the above pre-processing strategy. Therefore, we additionally defined and identified "peak regions", genomic regions in which 3' ends of mapped reads pile up, and calculated separate count statistics for each such peak (Section 4 in S1 Appendix).

In our experiment, observed T>C conversions in a sequence read can be caused on purpose by 4sU incorporation into newly synthesized RNA, followed by IAA conversion. But T>C conversions can also be artefacts resulting from SNPs in our HeLa cell line compared to the hg19 reference genome, or they can be caused by RNA editing. Lastly, such conversions can be the result of reverse transcription and sequencing errors. While the latter errors are stochastic and must be accounted for later in an error model, the first two types of errors are systematic (they introduce bias) and can be identified and corrected. We excluded all known potential SNP sites reported in the hg19 NCBI dbSNP database [62]. Additionally, all T sites with an observed conversion rate higher than 5% in the nuclear or cytosolic fraction of the unlabeled control sample were considered putative SNPs or editing sites and masked. Bias correction in the control sample, particularly removal of RNA editing sites, resulted in more similar single nucleotide mismatch rates between both strand-specific A>G or T>C and other conversions, between the nuclear and cytosolic fractions, and between 3'UTR and non-3'UTR peaks, indicating that our correction was effective (Section 8 and Fig M in S1 Appendix). SNP and RNA editing site correction was then applied to all samples of the two labeling time series (see Section 5 and Figs N-O in S1 Appendix for T>C conversion and mismatch statistics).

## Estimation of new/total mRNA ratios

The relative of abundances of newly synthesized among all transcripts of a population—henceforth called new/total ratios—are the endpoints predicted by our two-compartment model (Eqs (3) and (4)). The empirical labeled/total RNA ratio is a bad estimate of the new/total ratio, as newly synthesized transcripts might escape labeling due to low 4sU incorporation rates (2% [29]). It is common to use a binomial mixture model to estimate new/total ratios from the observed labeled and unlabeled reads [26, 35]. We use a slightly modified method for the estimation of the new/total ratios.

Fix a 3'UTR respectively a peak region onto which $J$ reads were mapped. Given a read $j$, let $T_j$ be its number of T-positions in the genomic sequence of the read alignment, and let $o_j$ be the number of T>C conversions observed in that read. Let $h_j \in \{0, 1\}$ be a hidden variable indicating whether read $j$ originates from a pre-existing RNA ($h_j = 0$) or a newly synthesized RNA ($h_j = 1$). The target of inference is the fraction $\rho \in [0, 1]$ of the reads originating from a newly synthesized RNA. We assume that the probability of a T>C conversion is constant for

positions in all newly synthesized reads, i.e., T positions are labeled with a constant labeling efficiency $\ell \in [0, 1]$. We found that this assumption is invalid for short labeling pulses of a few minutes (Section 7 in S1 Appendix). Yet, for the labeling durations in our experiment, we may safely take $\ell$ as a constant. Further, let $p_0$ be the probability of a T>C sequencing error, and $\epsilon$ the probability of a C>non-C sequencing error. Both $p_0$ and $\epsilon$ are estimated from the control RNA samples (Section 6 in S1 Appendix). We model the probability of seing a T>C converted nucleotide in a read originating from a newly synthesized transcript as $p_1 = \ell(1 - \epsilon) + (1 - \ell)p_0$. Assuming the reads are drawn independently, and all T>C conversions and sequencing errors occur independently, the likelihood function becomes

$$P(o, h) \quad = \quad \prod_{j=1}^{J} p(o_j, h_j; \rho, \ell) = \prod_{j=1}^{J} P(h_j; \rho) \cdot P(o_j \mid h_j; \ell) \tag{5}$$

with

$$P(h_j; \rho) \quad = \quad \mathrm{Bernoulli}(h_j; \rho) \tag{6}$$

$$P(o_j \mid h_j; \ell) \quad = \quad \mathrm{Bin}(o_j; T_j, p_{h_j}) \tag{7}$$

In a first round, we estimate the unknown parameters $\rho$ and $\ell$ by a standard EM algorithm, separately for each 3'UTR respectively peak region $g$ (Section 6 in S1 Appendix). It has already been noted in [26] that such a model does not produce sensible estimates whenever $\rho_g$ is small (which occurs particularly often for short labeling intervals). The reason is that it is advantageous to use both binomial mixture components for the modeling of the pre-existing transcripts than fitting one mixture component to the newly synthesized transcripts. Juerges et al. (2018) [26] address this problem by filtering out reads with a low number of T>C conversions. This however leads to highly variable numbers of reads per sample (a few thousand up to more than $10^5$) that enter the EM algorithm as input data and is therefore potentially susceptible to biases. To cirumvent this problem, we remove all regions $g$ for which the labeling efficiency was estimated below 1% and that are suspicious of the above overfitting phenomenon. Note that this means we generally consider samples with labeling efficiencies below 1% unsuitable for estimating new/total ratios. Fortunately, this is not the case in any of our samples. We then fix $\ell$ to the median of the remaining regions in a sample and calculate the maximum likelihood, region-specific estimates $\rho$ separately for each region $g$ (Section 6 in S1 Appendix).

We would like to point out that all models for estimating the new/total ratio known to us, including the one presented here, neglect some effects that lead to a delayed aggregation of labeled transcripts. First, the time it takes to add a polyA-tail to an mRNA after transcription termination is an interval during which newly synthesized mRNA is not detected by our method [35]. The effective labeling time is therefore the time after 4sU addition, reduced by this "detection gap". On the other hand, variations in the duration of labeling periods or delays in cell harvesting/processing can shift the offset in a negative direction. Because newly synthesized transcripts are produced continuously, their maturation time is highly variable. Consequently, the labeling efficiency with which their respective 3'-ends were synthesized cannot be assumed to be identical for all newly synthesized transcripts within one sample. We have extensively investigated these potential sources of bias to verify that they are negligible here, although this is not the case for short labeling pulses <5min (Section 7 in S1 Appendix).

## Fitting of metabolic parameters

Fix a transcript for which we want to fit the transcript-specific parameters $\Theta = (\tau + v, \lambda)$. Solving Eqs (1) and (2) of the main text with initial conditions $N(0) = 0$, $C(0) = 0$ and the parameters $\Theta$, the abundance of the newly synthesized RNA fraction in the nucleus and the cytosol is

$$N(t) = N_\infty \cdot (1 - e^{-(v+\tau)t})$$

$$C(t) = C_\infty \cdot \left(1 - \frac{\lambda e^{-(v+\tau)t} - (v+\tau)e^{-\lambda t}}{\lambda - (v+\tau)}\right)$$

where $N_\infty = \frac{\mu}{v+\tau}$ and $C_\infty = \frac{\tau}{\lambda} N_\infty$. The relative abundances ($\frac{new}{total}$) ratios are then given by $n(t) = \frac{N(t)}{N_\infty}$ and $c(t) = \frac{C(t)}{C_\infty}$. This yields a series of predictions of the nuclear and cytosolic new/total ratios $n_i = n(t_i)$ and $c_i = c(t_i)$, $i = 1,\ldots, T$, where $t_i$ are the time points at which observations were made. A straightforward approach is to fit a binomial model of the newly synthesized nuclear and cytosolic reads with the respective total number of reads, $R_i^{\mathrm{nuc}}$ and $R_i^{\mathrm{cyt}}$, and the ratios $n_i$ and $c_i$ as parameters. However, this fit is not robust against outliers and slight violations of the model assumptions. Alternatively, one could perform a least squares fit of the nuclear and cytosolic new/total ratios $\rho_i^{\mathrm{nuc}}$ and $\rho_i^{\mathrm{cyt}}$ which have been estimated above at each of the observation time points $t_i$, $i = 1,\ldots, T$:

$$\ell(\Theta) = \sum_{i=1}^{T}(\rho_i^{\mathrm{nuc}} - n_i)^2 + \sum_{i=1}^{T}(\rho_i^{\mathrm{cyt}} - c_i)^2$$

However, this does not take into account the unequal variance of the individual summands, which varies largely with $\rho_i^{\mathrm{nuc}}$ respectively $\rho_i^{\mathrm{cyt}}$. We therefore chose to combine both approaches and apply a variance stabilizing transformation. It was demonstrated that for sufficiently large $R$, a binomial variable $N \sim \mathrm{Binom}(R, p)$, after transformation $N \mapsto \sqrt{4R} \cdot \arcsin\sqrt{\frac{N}{R}}$, can be approximated by a Gaussian variable $Y \sim \mathcal{N}(\mu, \sigma^2)$ with $\mu = \sqrt{4R} \cdot \arcsin\sqrt{p}$ and $\sigma^2 = 1$ [63]. This approximation has been shown to be more accurate than the approximation of $N$ by a Gaussian with mean $p$ and variance $Rp(1 - p)$. Importantly, the variance of the transformed variable is approximately independent of its mean, i.e., the proposed transformation is variance stabilizing. We have verified the variance-stabilizing property of this transformation in our data (Fig L in S1 Appendix). We exploit this property and minimize the negative log likelihood of the Gaussian approximation, which is, up to additive and multiplicative constants,

$$\ell(\Theta) = \sum_{i=1}^{T} R_i^{\mathrm{nuc}}\left(\arcsin\sqrt{\rho_i^{\mathrm{nuc}}} - \arcsin\sqrt{n_i}\right)^2 + \sum_{i=1}^{T} R_i^{\mathrm{cyt}}\left(\arcsin\sqrt{\rho_i^{\mathrm{cyt}}} - \arcsin\sqrt{c_i}\right)^2$$

Instead of fitting $\Theta = (\tau + v, \lambda)$ jointly, parameter estimation was performed for $\tau + v$ first using the nuclear compartment model only. Then, the estimate of $\tau + v$ was plugged into the cytosolic compartment model for the fit of $\lambda$. The rationale behind this procedure is that a simple one-compartment is more stable to fit since the new/total ratios are higher in the nucleus. Further, according to our model, a variation in $\lambda$ does not affect the nuclear metabolism, and hence the estimate of $\tau + v$ should not depend on cytosolic observations.

Approximate confidence intervals for $\tau + v$ and $\lambda$ were calculated as the component-wise central 95% interval of an MCMC sample. To enhance convergence (burn-in), the MCMC was initialized with a maximizer of $\ell(\Theta)$ obtained by standard Nelder-Mead numerical optimization. Finally, a point estimate for $\Theta$ was obtained by another round of optimization starting

from the best scoring MCMC sample. Parameter estimation was executed separately for sample 1 and 2, and estimates were averaged afterwards.

## Reliability criteria

The metabolic rate estimate of a 3'UTR was classified as reliable if the following criteria were met: 1.) The 3'UTR had a minimum average read count of 30 in each the nuclear and the cytosolic fraction of each time series experiment. 2.) The 3'UTR's expression level was constant across the measured time series (steady state assumption): A simple linear regression was performed on the 3'UTR's expression level across each of the two time series. If the average slope of both fits did not exceed a value of 0.0025 (resp. not come below a value of -0.0025), the expression was considered constant. 3.) The relative deviation of the single time series' estimates, $x_1$ and $x_2$, was not higher than 33%, i.e. $\frac{2|x_1-x_2|}{x_1+x_2} \leq 0.33$. 4.) The relative deviation of the confidence interval's upper and lower bound, $c_u$ and $c_l$, was not higher than 0.3, i.e. $\left|\frac{2c_u}{x_1+x_2} - 1\right| \leq 0.3$ and $\left|\frac{2c_l}{x_1+x_2} - 1\right| \leq 0.3$. 5.) The rate estimate provided a reasonable fit to the arcsin-transformed new/total ratios ($R \geq 0.4$). Note that the cytosolic estimates were only considered reliable if the respective nuclear estimates were reliable as well, since the nuclear parameters were plugged into the cytosolic compartment model.

## Supporting information

**S1 Appendix. Supplementary information on method development.**
(PDF)

**S2 Appendix. Supplementary information on results.**
(PDF)

**S1 Table. RNA metabolic rate estimates.**
(CSV)

**S2 Table. List of supernova genes.**
(CSV)

**S3 Table. Annotation file for 3'UTR IDs and the corresponding Gene IDs.**
(CSV)

## Acknowledgments

The authors wish to thank Björn Schwalb for his valuable input on this subject. We thank Patrick Cramer for fruitful discussions about this project. We thank the Regional Computing Center of the University of Cologne (RRZK) for providing computing time and support on the DFG-funded (grant number: INST 216/512/1FUGG) High Performance Computing (HPC) system CHEOPS.

## Author Contributions

**Conceptualization:** Kristina Zumer, Achim Tresch.

**Data curation:** Jason M. Müller, Katharina Moos, Till Baar.

**Formal analysis:** Jason M. Müller, Katharina Moos.

**Funding acquisition:** Achim Tresch.

**Investigation:** Jason M. Müller, Katharina Moos, Kerstin C. Maier, Kristina Zumer, Achim Tresch.

**Methodology:** Jason M. Müller, Katharina Moos, Till Baar, Achim Tresch.

**Project administration:** Achim Tresch.

**Software:** Jason M. Müller, Katharina Moos.

**Supervision:** Kristina Zumer, Achim Tresch.

**Visualization:** Jason M. Müller, Katharina Moos.

**Writing – original draft:** Jason M. Müller, Jason M. Müller, Katharina Moos, Kristina Zumer, Achim Tresch.

**Writing – review & editing:** Kristina Zumer, Achim Tresch.

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
