## [Decision Letter · Decision Letter 0]

25 Jan 2024

Dear Prof. Tresch,

Thank you very much for submitting your manuscript "Nuclear export is a limiting factor in eukaryotic mRNA metabolism" for consideration at PLOS Computational Biology.

As with all papers reviewed by the journal, your manuscript was reviewed by members of the editorial board and by several independent reviewers. In light of the reviews (below this email), we would like to invite the resubmission of a significantly-revised version that takes into account the reviewers' comments.

We cannot make any decision about publication until we have seen the revised manuscript and your response to the reviewers' comments. Your revised manuscript is also likely to be sent to reviewers for further evaluation.

Sincerely,

Saurabh Sinha

Guest Editor

PLOS Computational Biology

Ruth Baker

Section Editor

PLOS Computational Biology

Reviewer's Responses to Questions

**Comments to the Authors:**

Reviewer #1: In this study, Mueller et al. performed SLAM-seq time series experiments that are based on metabolic labeling using the nucleotide analog 4-thiouridine (4sU). Following metabolic labeling cells were fractionated into nuclear and cytosolic fractions, and the nuclear and cytoplasmic RNAs are subjected to SLAM-seq library preparation and 3’-RNA-seq, obtaining subcellular resolution. The authors use the time course SLAM-seq data to develop a two-compartment model of RNA metabolism to quantify the key RNA metabolic parameters: the mRNA synthesis-, nuclear export-, nuclear- and cytosolic degradation-rates. The authors find that degradation of polyadenylated mRNA in the nucleus is negligible and that nuclear export is slow whereas the degradation of mRNA in the cytosol is relatively fast. From these findings the authors conclude that polyA(+) mRNAs generally spend most of their time in the cell nucleus and that RNA export is a rate limiting factor in the mRNA’s life cycle. The authors also provide evidence that the nuclear export rate can differ strongly for transcript isoforms with different 3’-UTRs suggesting that the 3’-UTR is an important determinant for nuclear export rates. Interestingly, the authors find that a subset of genes are activated upon metabolic labeling. This set of labeling-induced mRNAs exhibit higher nuclear export rates as other mRNAs which points to different mRNA export pathways.

The findings of this combined experimental and computational study are exciting. Of particular relevance, at least in my opinion, is the finding that polyadenylated mRNAs generally spend most of their life time in the nucleus. The methods that the authors have used are state-of-the-art. The experiments were well designed and the advanced computational analyses have been thoroughly performed. The study opens new questions that can be addressed in future studies. For instance, the data suggests the co-existence of alternative export pathways. The identification and characterization of these pathways represents an important future direction. The findings of this work will be of broad relevance for the large community of researchers with an interest in a systems-level understanding of the RNA life cycle and especially to those who work on mRNA metabolism and RNA export.

Here, some suggestions that the authors may want to consider to further improve this work.

Major comments:

(1) The introduction section of the manuscript should be better adapted to the main subject and results of this study. The authors should also emphasize what is not known helping the reader to understand the motivation and need of this systems-wide RNA analysis.

(2) The first paragraphs of the results section (pages 3 to 7) are very technical and appear as part of the methods section. I understand that the authors present a new model to quantify main parameters of the RNA life cycle and therefore technical aspects are important. However, I find especially the first paragraphs of the results section hard to read. The authors may also want to consider to reduce the load of formulas in the main text.

(3) On page 3 the authors mention that two replicate samples were obtained per time point. The authors should provide a correlation heatmap with the correlation coefficients between the replicates and between time points. This would further (in addition to Fig. 1D) demonstrate the reproducibility of the data.

Minor comments:

(1) The authors should mention the model system in the abstract: human cancer cells.

(2) A more detailed description of how 3’-UTR isoforms were defined in the methods section would be helpful. How many 3’-UTR isoforms were identified? How many per transcript?

(3) The authors nicely compare their obtained RNA half-lives (HeLa) with already available half-lives from MCF7 and HEK293 obtained in previous studies by others. The authors found that the half-live estimates from HEK293 and MCF7 cells were systematically shorter or longer than the half-lives obtained for HeLa cells (Page 7). What are potential reasons for the systematic differences? The authors could elaborate on this in the main text.

(4) Fig. 3B: text of the x-axis is partially overlapping

Reviewer #2: Muller et al. presented a dynamic model of time-resolved SLAM-seq data to quantify two key parameters of RNA metabolism, namely the export of mRNA from nuclear compartment to cytosol and degradation. It is a two-compartment model, which should capture the main steps of RNA metabolism (mRNA synthesis, nuclear export, and nuclear and cytosolic degradation rates) more accurately than existing models like GRAND-SLAM that assume constant RNA synthesis rates, degradation rates,

and a dynamic equilibrium of each RNA population. The manuscript’s main finding is that mRNA residence time in the nucleus is much longer than that in the cytosol. The authors also showed that different 3’UTR transcript isoforms can be exported at substantially different rates and that stress by 4sU labeling can induce alternative export pathways.

Overall, it is a well written paper with careful pre-processing of SLAM-seq data. Below we list our main criticisms, which primarily stem from overly generalized claims and unjustified assumptions.

1. The authors analyzed one dataset, but consistently made general biological statements. As presented, it is unclear why the authors did not use other published datasets (e.g., https://elifesciences.org/articles/45396) and if their statements will generally hold on to other datasets.

2. The authors clearly mentioned their assumptions but didn’t justify them well. For example, they noted: “Since our experiments are conducted under constant, optimal growth conditions, we assume these four rates are constant over time” and “we further assume that the metabolic parameters of 4sU-labeled RNA are identical to those of unlabeled RNA.” Both assumptions are questionable unless presented with prior evidence. It is also critical to assess (at least, to comment on) how these assumptions might affect their conclusions.

3. The assumption of global parameters for mRNA synthesis, export, and degradation is also questionable. Won’t the size of the dataset allow for at least binary categorizations (high/low and fast/slow) of these parameters?

4. Critical parts of the approach, such as estimation of new/total mRNA ratios, need to be validated on additional datasets. The authors seemed forceful at some places. For example, they noted in S5 that “Contrary to our intuition, the estimated value for the maturation gap was negative. We still keep our model, since it represents an excellent fit of labeling efficiency.”

5. Due to points 2-4, much of the presented results remain questionable.

6. The authors wrote a very nice Introduction section, but similar elaborations of the SLAM-seq technology would be helpful.

7. The github link (https://github.com/IMSBCompBio/mRNAdynamics) was not accessible.

**Have the authors made all data and (if applicable) computational code underlying the findings in their manuscript fully available?**

Reviewer #1: Yes

Reviewer #2: **No: **The github link (https://github.com/IMSBCompBio/mRNAdynamics) was not accessible.

PLOS authors have the option to publish the peer review history of their article (what does this mean?). If published, this will include your full peer review and any attached files.

Reviewer #1: No

Reviewer #2: No
---

## [Decision Letter · Decision Letter 1]

9 Apr 2024

Dear Prof. Tresch,

We are pleased to inform you that your manuscript 'Nuclear export is a limiting factor in eukaryotic mRNA metabolism' has been provisionally accepted for publication in PLOS Computational Biology.

Before your manuscript can be formally accepted you will need to complete some formatting changes, which you will receive in a follow up email. A member of our team will be in touch with a set of requests. In addition to these formatting changes, please make sure that the github link for your software, which is currently inaccessible, is accessible to the public. 

Best regards,

Saurabh Sinha

Guest Editor

PLOS Computational Biology

Pedro Mendes

Section Editor

PLOS Computational Biology

Reviewer's Responses to Questions

**Comments to the Authors:**

Reviewer #1: The authors have addressed all of my comments. Notably, the introduction is now well aligned with the main results of the study and better describes the motivation of this work. The first results paragraphs are strongly improved and the reproducibility of the data is clearly demonstrated, also by new correlation analyses (Figures S17 to S19). Overall, the authors have done an exceptional job.

Reviewer #2: The authors have satisfactorily addressed our major comments and concerns.

**Have the authors made all data and (if applicable) computational code underlying the findings in their manuscript fully available?**

Reviewer #1: Yes

Reviewer #2: **No: **The authors cited "intellectual property reasons" and didn't make the GitHub public yet.

PLOS authors have the option to publish the peer review history of their article (what does this mean?). If published, this will include your full peer review and any attached files.

Reviewer #1: No

Reviewer #2: No

---

## [Editor Report · Acceptance letter]

13 May 2024

PCOMPBIOL-D-23-01695R1 

Nuclear export is a limiting factor in eukaryotic mRNA metabolism

Dear Dr Tresch,

I am pleased to inform you that your manuscript has been formally accepted for publication in PLOS Computational Biology. Your manuscript is now with our production department and you will be notified of the publication date in due course.

With kind regards,

Lilla Horvath
